# Microbiota Influences Fitness and Timing of Reproduction in the Fruit Fly *Drosophila melanogaster*

Melinda K. Matthews,[a] Jaanna Malcolm,[a] John M. Chaston[a]

[a]Department of Plant and Wildlife Sciences, Brigham Young University, Provo, Utah, USA

**ABSTRACT** Associated microorganisms ("microbiota") play a central role in determining many animals' survival and reproduction characteristics. The impact of these microbial influences on an animal's fitness, or population growth, in a given environment has not been defined as clearly. We focused on microbiota-dependent host fitness by measuring life span and fecundity in *Drosophila melanogaster* fruit flies reared individually with 14 different bacterial species. Consistent with previous observations, the different bacteria significantly influenced the timing of fly life span and fecundity. Using Leslie matrices, we show that fly fitness was lowest when the microbes caused the flies to invest in life span over fecundity. Computational permutations showed that the positive fitness effect of investing in reproduction was reversed if fly survival over time was low, indicating that the observed fitness influences of the microbes could be context dependent. Finally, we showed that fly fitness is not influenced by bacterial genes that shape fly life span or fly triglyceride content, a trait that is related to fly survival and reproduction. Also, metagenome-wide association did not identify any microbial genes that were associated with variation in fly fitness. Therefore, the bacterial genetic basis for influencing fly fitness remains unknown. We conclude that bacteria influence a fly's reproductive timing more than total reproductive output and that (e.g., environmental) conditions that influence fly survival likely determine which bacteria benefit fly fitness.

**IMPORTANCE** The ability of associated microorganisms ("microbiota") to influence animal life history traits has been recognized and investigated, especially in the past 2 decades. For many microbial communities, there is not always a clear definition of whether the microbiota or its members are beneficial, pathogenic, or relatively neutral to their hosts' fitness. In this study, we report the influence of individual members of the microbiota on *Drosophila melanogaster* fitness using Leslie matrices that combine the microbial influences on fly survival and reproduction into a single fitness measure. Our results are consistent with a previous report that, in the laboratory, acetic acid bacteria are more beneficial to the flies than many strains of lactic acid bacteria. We add to the previous finding by showing that this benefit depends on fly survival rate. Together, our work helps to show how the microbiota of a fly influences its laboratory fitness and how these effects may translate to a wild setting.

**KEYWORDS** acetic acid bacteria, lactic acid bacteria, Leslie matrix, fitness, *Drosophila melanogaster*, microbiota

Address correspondence to John M. Chaston, john_chaston@byu.edu.

Organismal fitness, or population growth, is defined by the relationship between an organism's survival and reproduction. The most fit organism would live forever and have an infinite number of offspring each day, but biological limitations and resource availability constrain this hypothetical maximum. Instead, organisms must invest their available resources between survival and reproduction. The relationship between the two traits often occurs as a trade-off, where resource investment in

survival traits comes at some cost to reproductive output (1–4). Differential resource investment to survival and reproduction often occurs within members of the same species that live in different geographic areas as a result of local adaptation (1, 5). However, the trade-off is not essential for local adaptation (6), and exceptions to the trade-off have been documented when an animal's genotype is modified or under specific selection regimes (7–10). Where the trade-off is detected, the preferred resource investment outcome is often described as being along the fast-slow continuum, where investment in somatic maintenance ("survival") traits is considered "slow" and greater resource allocation to reproduction is called "fast" (4, 11–13). Organisms that invest all resources in either reproduction or somatic maintenance are located at the hypothetical extrema of the continuum, and organisms with a more balanced trade-off classified closer to the center. Recently, we showed that an isogenic animal host colonized with different bacteria varied in its position along the fast-slow continuum (14). However, we did not report if the host's fitness was superior when reared with reproduction- or survival-maximizing microbes. The work presented here defines the fitness consequences to the host of microbial influences that change the host's position on the fast-slow continuum. Throughout this work we estimated reproduction by measuring fly fecundity and defined survival as fly life span under laboratory conditions at specified intervals.

The fruit fly *Drosophila melanogaster* has emerged over the last decade as a powerful tool for understanding the ecology of animal-microbiota interactions. The microbiota of wild and laboratory fruit flies is of low taxonomic and numerical diversity relative to most mammals, and the most common and abundant microbes are the acetic acid and lactic acid bacteria (AAB and LAB, respectively) and *Enterobacteriaceae*. (15–20). Most relevant to fitness, previous work reported the varied influences of single- and polyspecies microbial communities on fly fitness, with maximum fitness being observed in flies that bore AAB abundantly; these bacteria promoted a "fast" high-reproduction lifestyle in the flies (21). Several gaps remain for further investigation into the fitness influences of the microbiota on *D. melanogaster*. For example, the previous experiments measured adult fecundity in flies that were reared bacteria-free to 8 days of age and then associated with different microorganisms. While this approach normalized for the different influences of the bacteria on fly development, bacteria-dependent influences on timing of and/or prior to reproductive maturity are also important considerations in the microbial influence on fitness. Thus, the current description of how microbial colonization influences *Drosophila melanogaster* fitness is incomplete.

To better understand how fly fitness relates to changes in microbiota composition, we studied the relationship between reproduction and survival in flies reared with individual microorganisms. This goal was motivated by our detection of a geographic pattern in the microbiota composition of wild flies, where AAB and LAB were more abundant in flies at lower and higher latitudes, respectively, in multiple locations, and work showing that such variation in microbiota composition was likely linked to the adaptive traits of the flies (14). However, the previous work did not address variation in fly fitness. For congruence with that previous work and to understand the roles of individual microbes, we used monoassociated animals for these experiments and asked three questions. (i) How does fitness vary in flies that are reared lifelong with different microorganisms? (ii) Is fly reproduction or survival more important to microbe-dependent fitness? (iii) Do bacterial genes that influence individual fly life history traits also shape fly fitness? To answer these questions, we measured fecundity and longevity as proxies for fitness in flies reared from birth with each of 14 different microorganisms. The strains that we used were selected to obtain a broad representation of AAB and LAB, with several additional strains to add genetic diversity to the strain panel. We then calculated fly fitness using Leslie matrices, including the effect of simulating decreased reproductive output or survival in specific time intervals. Finally, we measured fitness of flies reared with specific bacterial mutants and conducted a metagenome-wide association (MGWA) study to identify putative bacterial determinants of fly fitness. Our

Microbiology Spectrum

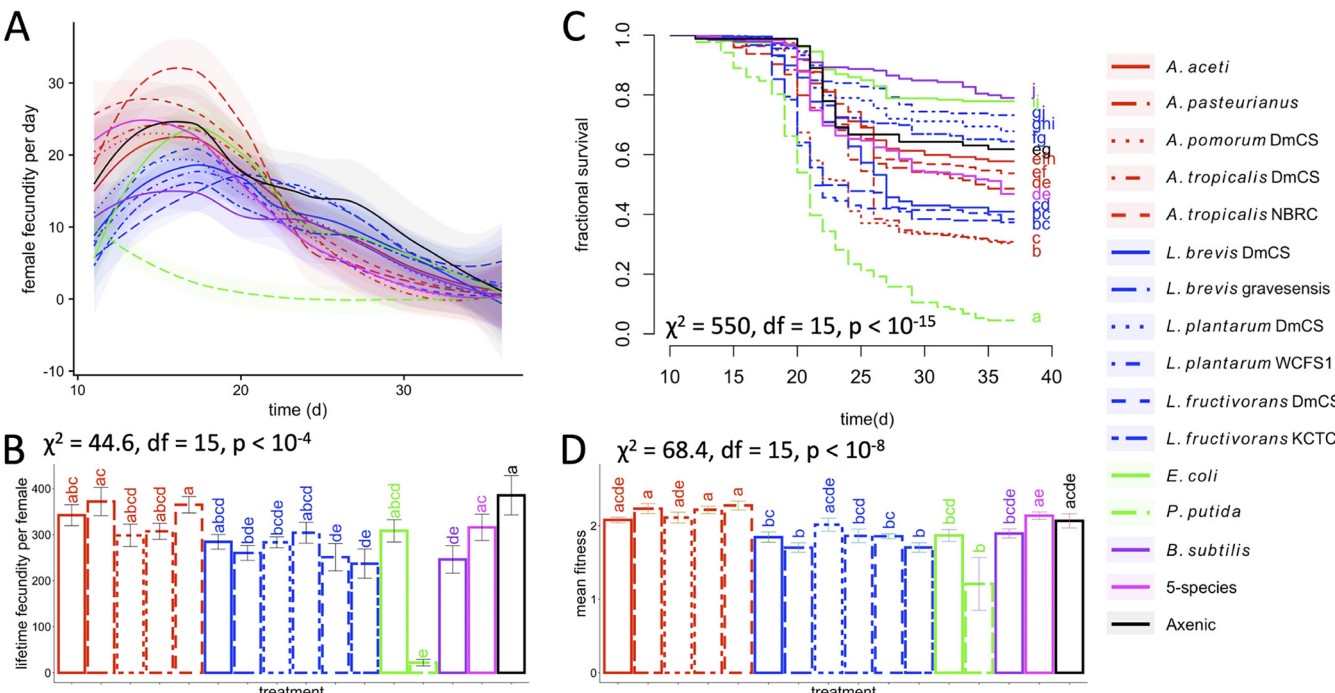

**FIG 1** Natural microbiota isolates influence *D. melanogaster* fecundity, life span, and fitness. (A) For each of the microbial treatments, the average number of offspring produced per *D. melanogaster* female each day, measured in three time-staggered experiments each with twice weekly intervals across the life cycle. (B) The average total estimated offspring produced per female. Significant differences between treatments were defined by a Kruskal-Wallis test. (C) Fly life span when reared with the same microorganisms as in the previous panels, and significant differences were determined by a log rank test. (D) The effect of fly survival during the first reproductive period on fitness lambda, calculated as the first eigenvalue from a Leslie matrix. Colors are distributed according to the high-level taxonomic classification of the microbial treatment as follows: red, AAB; blue, LAB; purple, *Bacillus subtilis*; green, *Enterobacteriales*; magenta, 5-species; black, axenic. Significant differences between treatments were determined by a Kruskal-Wallis test. In all panels, significant differences between treatments are shown by compact letter displays.

results indicated that AAB provided the greatest fitness benefit to the flies, an effect that depended on relatively high survival rates of the flies during early life stages. However, we were not able to identify any bacterial genes that influenced fly fitness.

## RESULTS

**The microbiota influences *D. melanogaster* reproductive timing and output.** To define microbial influences on the timing of *D. melanogaster* reproductive output, we tracked fecundity in sequential windows of time, using flies that were bacteria-free or colonized individually with 14 different bacterial species (Fig. 1A). One trend was that the bacteria had different effects on the timing of fly fecundity: at early and late time points, fecundities were highest in flies colonized by the AAB or LAB, respectively. The average timing of peak fly fecundity across the different treatments spanned 8.5 days, ranging from 12.5 days for *Pseudomonas putida* F1 to 21 days for *Lactobacillus brevis* subsp. *gravesensis* ATCC 27305 (see Table S1 in the supplemental material). Unlike for fecundity, the microbial influence on fly development time to adulthood varies over less than 24 h (e.g., reference 22, which uses the same bacterial strains, fly genotype, and diet as this work), indicating that bacterial effects on fly development cannot be the only cause of the variation in peak fly fecundity. The microbial inoculations also influenced lifelong fecundity of the flies (Fig. 1B), but many AAB and LAB conferred similar values for this trait, suggesting that the microbes had a stronger influence on the timing of fecundity than on overall reproductive output. Consistent with reports from our previous experiments with the same fly and bacterial strains and the same diet (14), individual bacteria that conferred higher early adult fecundities usually led to shorter fly life span (compare to Fig. 1C with survival statistics in Table S2 in the supplemental material, for survival measured from the same fly vials as the P generation in the fecundity

experiments). The life span data also provided insight into some of the fecundity results. For example, *Pseudomonas putida* F1 led to short fly life span, suggesting its negative effects on fly fecundity may have been tied to a general deleterious or pathogenic influence on the flies. Together, these results show a more dramatic influence of the microbes on timing of, relative to total, reproductive output of the flies, with a general trend that LAB-inoculated flies displayed a trade-off that reduced early fecundity in favor of life span and late fecundity as we documented previously (14).

**The microbiota influences *D. melanogaster* fitness.** To better understand the relationship between microbially influenced life span and timing of fecundity on the overall fitness of the flies, we calculated fly fitness as the eigenvalue lambda of a Leslie matrix. Leslie matrices are age-structured models of population growth where both fly fecundity and survival are measured in discrete intervals or stages (23). We defined the age classes in twice-weekly intervals and measured the fecundity of a mixed-sex fly population at each interval and fly survival rates between intervals. When we calculated fitness lambda under these conditions, we detected that the different bacterial treatments led to significant variation in fly fitness. Also, bacteria that conferred the highest early fecundity tended to confer the highest fitness values (Fig. 1D). These findings confirmed that, consistent with reports from a different laboratory, the varying influences of the different microorganisms on *D. melanogaster* fecundity and life span led to distinct fly fitness outcomes (21). Another similarity between our work and Gould et al. (21) was that in monoassociation the AAB tended to confer higher fitness values on flies than LAB (21). Thus, these findings are consistent with a previous report and, by testing a broader bacterial strain panel, provide stronger support for the observation that AAB are more beneficial to the fitness of laboratory flies than are LAB isolates.

**Survival of the first *D. melanogaster* age class determines the bacterial fitness benefit.** The conclusion that AAB can benefit fly fitness more than LAB raises the following question: why do wild flies at some geographic locations naturally bear and genetically select for greater abundances of LAB than flies at other locations (14)? Fly genotype likely plays a role, but in our experiments, we only tested one fly genotype. We reasoned that geography might also be important if fly survival and timing of fecundity have different importance in distinct locations. For example, if flies in one location have poorer survival than flies in another location, the relative fly fitness benefits of the bacterial strains might vary. We investigated this idea by simulating changes in fecundity and survival in the flies using computational modeling, each change applied as a constant across all treatments, in different fly age classes. For example, to test the contributions of the bacterial impact on fly survival to fitness during the youngest age class, which corresponded to 0- to 3-day-old adults, we (i) sequentially simulated reductions in fly survival by factors of 2, 10, 20, 100, 200, 1,000, 10,000, and 100,000; (ii) calculated fitness based on the permuted survival values; and (iii) compared the fly fitness values conferred by the different microorganisms by a Kruskal-Wallis test (see Fig. S1 in the supplemental material). We reached three conclusions. First, of all age classes, the youngest influenced fly fitness the most; with only a few exceptions, fitness was only altered significantly when survival or fecundity was permuted in this age class (see Fig. S1 and S2 in the supplemental material). Second, fly survival determined fly fitness more than fly fecundity did. This was apparent as the fitness values dropped more precipitously with a similar denominator change in survival than in fecundity (compare panels between Fig. S1 and S2). Third, fly fitness did not vary linearly with changes in fly survival. For example, the significant decrease in fitness for flies bearing *Lactobacillus plantarum* DmCS_002 relative to *Acetobacter pasteurianus* NBRC 101655 was reversed when fly survival was reduced by 5 orders of magnitude in the first age class (see Fig. S1). This effect was age class specific since a similar reduction in survival of other age classes did not cause the difference between these strains to change (Fig. S1). The most striking effect of these changes was apparent by comparing the fitness of each individual vial for the observed and permuted survival and fecundity values.

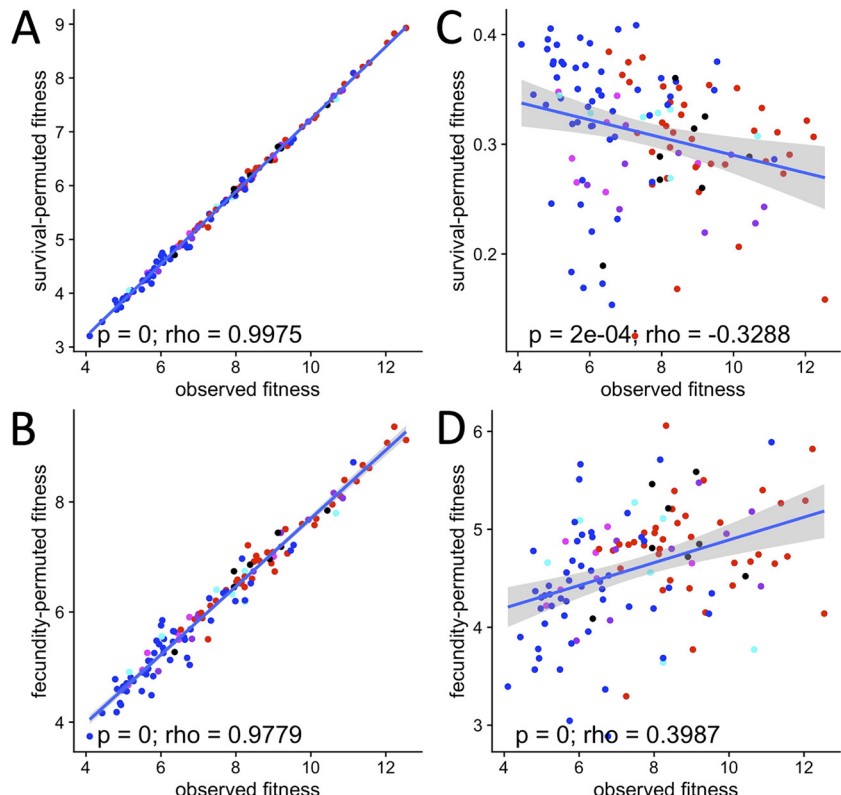

**FIG 2** Changes in *D. melanogaster* survival and fecundity differentially affect fly fitness. The relative contributions of microbiota influences on *D. melanogaster* fitness were determined by permuting fly survival (A, C) and fecundity (B, D) to 0.5 (A, B) or 0.00001 (C, D) of observed values. Each point represents a single vial, and values were permuted on a per-vial basis. Points are shaded according to the high-level taxonomic classification of the microbial treatment the vial received as follows: red, AAB; blue, LAB; magenta, *B. subtilis*; cyan, *Enterobacteriales*; purple, 5-species; black, axenic. The correlation between observed and permuted fitness values was calculated. *P*, *P* value; rho, Spearman's rho. Data from vials containing flies reared with *P. putida*, which had characteristics of a fly pathogen, were omitted from these analyses.

When survival or fecundity decreased by relatively small margins, observed and permuted fitness were positively correlated regardless of the age group or trait (survival or fecundity) that was modified (Fig. 2A and B; see also Fig. S3 and S4 in the supplemental material). However, large changes in survival eliminated or reversed the positive correlation between observed and permuted fitness (Fig. 2C). For example, at the observed survival rates shown in Fig. 1C, the highest average fitness was conferred by AAB strains such as *Acetobacter tropicalis* NBRC 101654 and *A. pasteurianus* NBRC 101655; however, at the lowest simulated survival rates, the highest average fly fitness was conferred by *L. plantarum* WCFS1 (Fig. S1). Thus, the relative fitness benefit of the different bacterial strains depended on the survival rate of the flies. Simulating lower fly fecundity did not reverse the fitness benefit of different bacterial strains (Fig. 2D; see also Fig. S2). The results of these computational simulations suggest that there may be age-class dependent influences of the microbiota on fly survival that help to shape context- and microbiota-dependent variation in fly fitness.

**Bacterial genes that influence fly life history traits do not influence fly fitness.** To better understand the bacterial-genetic basis for influencing *D. melanogaster* fitness, we adopted two parallel approaches. First, we tested if bacterial genes that influence fly life history traits also influence fly fitness. Fly triglyceride content and starvation resistance are positively correlated with host life span, including in response to the microbiota (14). Currently, the microbiota is understood to influence host traits such as these with or without dietary modification. For example, in *Acetobacter pasteurianus* 3p3, overexpression of *S*-oxidoreductase, gluconate dehydrogenase, and glucose

dehydrogenase genes led to increased catabolism of dietary glucose by the bacteria and lowered availability of dietary glucose for fly fat storage (22) (strains Sdr, Gndh, and Gdh from Table 2, respectively). The mechanism of influence for an *Escherichia coli* BW25113 *metH* mutation is unknown but is likely to be more precise than a major change in the composition of the diet because mutations in other methionine cycle genes do not substantially alter the metabolite content of the fly diet (24). We tested each of these four genes for influence on life span, fecundity, and fitness. For triglyceride content, the gluconate dehydrogenase overexpression strain reduced fly life span, which, combined with its known effect to reduce triglyceride content of the flies (22), is consistent with the idea that variation in life span and fat content traits is often positively correlated in wild *Drosophila* populations or with variation in the *Drosophila* microbiota (Fig. 3A) (14, 25). However, this strain did not affect the lifetime fecundity or fitness of the flies (Fig. 3B to D; see also survival statistics in Table S2). Also, neither of the glucose dehydrogenase or *S*-oxidoreductase overexpression strains, which also reduce fly triglyceride content, affected any of the traits that we measured (Fig. 3). Similarly, an *metH* mutant that affects *Drosophila* starvation resistance (26) but not life span (24) had no effect in this study on life span (raw statistics in Table S2), total fecundity, or fitness (Fig. 3). Finally, permuting survival up to 6 orders of magnitude did not cause differences in the fitness conferred by the strains (see Fig. S5 in the supplemental material). Therefore, the general finding from these experiments is that bacterial mutants that influence one life history trait do not necessarily influence other life history traits. These incongruent effects of the same bacterial mutants on different fly life history traits are also apparent between two of our previous studies that profiled the influences of an overlapping set of *E. coli* mutants on fly life span (24) and starvation resistance (26).

As a second approach to identify bacterial genes that influence host fitness, we performed a metagenome-wide association (MGWA). A total of 7,755 orthologous groups (OGs) were present in the whole-genome sequences of the exact strains that we tested in this study and were spread across 431 unique phylogenetic distribution groups (PDGs). Of these OGs and PDGs, none were significantly associated with fly fitness after correcting for multiple tests, and only 7 OGs in 4 PDGs were significantly associated with *D. melanogaster* fitness before such correction (Table 1; see also Table S3 in the supplemental material). These *P* values were deflated and yielded less statistical insight than previous MGWAs that we have performed, likely from lack of statistical power and genetic diversity due to use of a relatively small set of individual strains. Together, the two approaches were insufficient to confirm the identity of any bacterial genes that influence host fitness. However, some of the most significant MGWA-predicted candidate genes have links to host-microbe interactions and may be candidates for future study.

## DISCUSSION

In this work, we measured the influence of a broad set of microorganisms on the paired life spans and fecundity of *D. melanogaster* flies. The major observations were that the different bacteria significantly influenced fly fitness, life span, and the timing of fly reproductive output. Bacterial treatments also led to some changes in total fly fecundity. Additionally, simulated changes in fly survival rates during specified intervals suggested that the fitness benefit of the microbial partners might be more dependent on fly survival rates than fly reproductive output. Our efforts to define specific bacterial genes underlying these fitness influences were unsuccessful either by testing specific genes that influence individual life history traits or by performing MGWA. Overall, the results point to the important influence of associated microorganisms underlying host survival and reproduction phenotypes.

Different outcomes of our experiments overlapped with or varied from a previous report by Gould et al. (21) on the influence of bacterial colonization on fly fecundity, life span, and fitness. At least five major experimental differences between our study

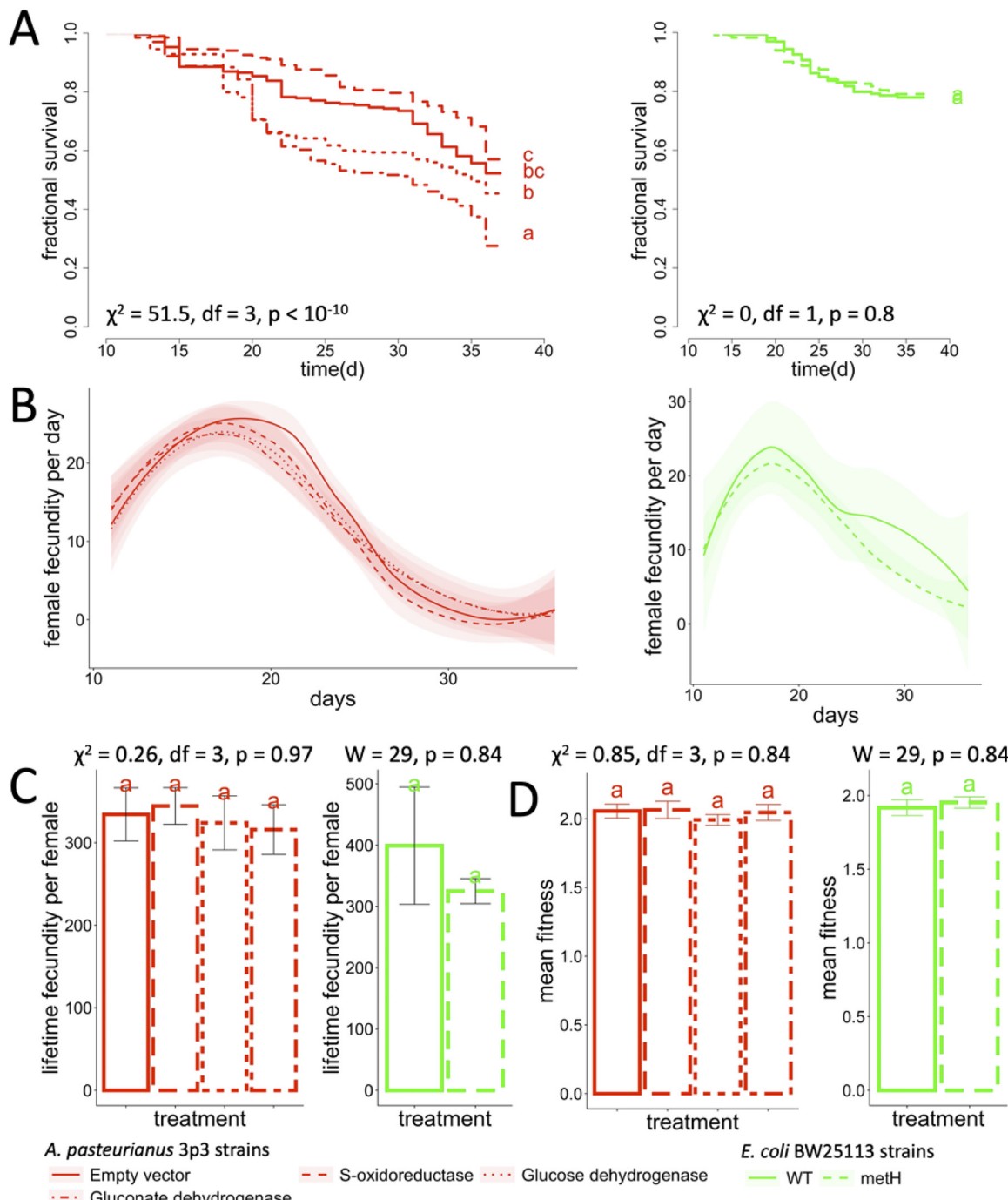

**FIG 3** Bacterial mutants that influence *Drosophila* life history traits do not influence fly fitness. Lifespan (A), daily fecundity (B), total fecundity (C), and fitness (D) of flies reared with bacterial mutants or control strains. Significant differences were determined by a log-rank test (A) or a Kruskal-Wallis (*Acetobacter* strains) or Wilcoxon (*E. coli* strains) test (C, D). Compact letter displays show significant differences between treatments. Strains are shown in red (*A. pasteurianus* 3p3 bearing plasmids pCM62 [empty vector], pCM62-SDR [*S*-oxidoreductase], pCM62-GDH [glucose dehydrogenase], and pCM62-GnDH [gluconate dehydrogenase]) or green (*E. coli* BW25113 [WT, wild type] and *E. coli* BW25113 Δ*metH*786::*kan* [metH]).

and that of Gould et al. included the following: the starting age of the flies, the fly diet, the method of generating defined bacterial associations, the density of flies in vials, and the method of populating the Leslie matrix. Each of these factors might reasonably be expected to influence fly reproductive output or survival, so it is not surprising that there were some differences in outcomes. One of the most striking differences in

**TABLE 1** Metagenome-wide association study

| P value[a] | Mean fitness (+)[b] | Mean fitness (−)[c] | Strains that contain the PDG[d] | Annotations of predicted genes |
|---|---|---|---|---|
| 0.02 | 1.91 | 2.12 | 5 LAB, 2 AAB, E. coli, B. subtilis | MerR family transcriptional regulator |
| 0.03 | 1.97 | 2.01 | 5 LAB, 3 AAB, E. coli, B. subtilis | DUF72 domain-containing protein |
| 0.03 | 1.78 | 2.01 | 2 LAB | Phage tail family protein; hypothetical protein; glycosyl hydrolase |
| 0.05 | 1.87 | 2.01 | 2 LAB, 1 AAB | Type II CRISPR-associated endonuclease Cas1; CRISPR-associated endonuclease Cas2 |

[a]MGWA P value (not corrected for multiple tests).
[b]Mean fitness of bacteria that contain the PDG.
[c]Mean fitness of bacteria that do not contain the PDG.
[d]Identities of strains that contain the PDG (of 5 AAB, 6 LAB, E. coli, and B. subtilis).

outcome was that our experiments captured increases in daily fly fecundity during early adulthood, whereas the Gould et al. experiments began at or near the peak daily reproductive output. Also, in Gould et al., the three AAB treatments displayed consistently higher lifelong fecundity than LAB treatments, whereas in our work, there was a trade-off between fecundity at high and late time points between treatments. This effect may have resulted from diet or other experimental differences. For example, relative to that of Gould et al., the diet in our analysis contained the same amount of sugar and ~44% more calories, including 179% more protein, 14% more fat, and 19% more carbohydrates (data from the Drosophila diet composition calculator [27]). Finally, unlike Gould et al., we did not measure bacterial abundance to test how it was related to fitness because we did not collect data for any polyassociated flies and because bacterial abundance is not linked to life span effects in monoassociated flies for the host genotype and bacterial strains in this study (24). Despite these differences between studies, the general outcomes were similar. AAB-colonized flies tended to confer greater total fecundity on the flies than LAB-colonized flies (Fig. 1B), and most treatments led to reproductive senescence around the same time period, rather than having a wide time-range when flies no longer laid viable eggs. By measuring the traits of flies colonized with a broader panel of additional isolates than in Gould et al., our study also shows that these traits are influenced by bacteria that are not normally detected abundantly in wild flies and supports the trend in Gould et al. of a bifurcation in the traits conferred by the LAB or the AAB. Together, both studies confirm that individual microorganisms can significantly alter reproductive timing and output, with consequent influences on fruit fly fitness.

In computer simulations, the benefits of the different bacteria to fly fitness depended on the survival rates of the flies. We hypothesize that a primary role for survival on fly fitness could be a clue to geography-specific selection patterns on the flies (14). For example, we can generalize the relative host fitness benefits of AAB versus LAB based on the context of fly survival and the geographic locations where those groups are the most abundant. Under actual survival in our experiments, AAB conferred higher average fitness on the flies than LAB (sometimes significantly so, sometimes not); however, under low simulated survival, some LAB strains conferred higher fitness than some AAB. Previous work showed that the ratio of AAB/LAB decreases with latitude in the eastern United States (14). Bringing these two pieces together enables us to speculate on at least two processes that could drive divergence in host genetic selection for LAB and AAB. First, if predation or death are major drivers of selection, then the greater relative abundance of AAB in flies in the southeast (SE) versus northeast (NE) United States may suggest SE flies naturally live longer in the wild than NE flies. Second, if selection is based on environmental influences on fly life history traits (higher temperatures and AAB colonization accelerate development to reproductive maturity [28] and increase initial fecundity [29, 30] relative to lower temperatures and LAB colonization), then congruence between microbial and environmental influences on life history traits promotes maximal fitness. These interpretations are based on simulations, lack confirmatory experiments, and are subject to caveats. Our experiments were performed as monoassociations on a single fly genotype in the

laboratory. How the patterns may change in polycolonized flies was investigated previously (21), is likely to depend on host genotype (e.g., references 14 and 31), and may be different in the wild than in the laboratory. Additionally, the computer simulations may not accurately reflect outcomes for flies in wild conditions. The phenotypes conferred by individual or pairs of bacteria sometimes can but commonly cannot predict the traits of flies colonized with multiple bacteria (21), meaning that monoassociation data cannot capture the full picture. However, in nearly all of the associations in a previous report, LAB were <10% of the microbial community (21). Therefore, the fitness and life history outcomes are unknown when flies are colonized with substantial relative abundances of LAB. Comparing traits between monoassociated flies, despite limitations, is a first step in understanding how variation in microbiota composition influences fly fitness. We are not aware of any techniques that establish different stable abundances of the same microbial community members without covarying key confounding factors (e.g., diet, host genotype, microbial taxonomy, maybe temperature [32]), but a systematic comparison of fly traits and microbiota while covarying such factors is a likely next step in understanding the relationship between microbiota composition and fitness of locally adapted fruit flies.

Our study supports the idea that bacterial genes that contribute to one fly life history trait do not necessarily influence other related life history traits. In wild fruit fly populations, trait values for fruit fly starvation resistance, triglyceride content, and life span are commonly positively correlated with each other and negatively correlated with development rate and early fecundity (25). Manipulation of fly genotype or diet can in some cases alter a specific, instead of all, correlated trait (9, 10, 33–37), showing that the traits are not controlled by a single underlying mechanism. Our work here adds to previous evidence that bacterial genes can also influence life history traits individually (24, 26) by showing that, under our conditions, bacterial genes that influence fly triglyceride content or starvation resistance do not influence fly life span, reproduction, and fitness. Thus, the bacterial genetic basis for influencing host traits is likely specific, at least in some instances, to the trait of interest. Alternatively, or additionally, these genetic influences may be context dependent, for example, with the presence of other microorganisms, dietary components, or environmental signals.

The MGWA that we performed identified no PDGs that were significantly associated with fly fitness after Bonferroni correction for multiple tests. Even before correction, only 7 OGs in 4 PDGs were associated with fly fitness. This is a much smaller list of candidate OGs than we have obtained in previous MGWAs (22, 24, 26, 38), likely because we collected experimental data for about 1/3 of the usual complement of strains as our previous studies. The smaller number of bacterial treatments was necessary to maintain a reasonable experimental size. The relatively small number of replicated measures also may have contributed to the low *P* values (7 to 9 replicates per treatment, based on triplicate measures in three separate experiments, with some replicates discarded due to bacterial contamination in the vials). Of the seven OGs that were significant before correcting for multiple tests, there are genes known to be involved in processes associated with host-microbe interactions, including a *merR* transcriptional regulator (39), CRISPR endonucleases Cas1 and Cas2 (e.g., reference 40), and energy utilization genes. We expect that the current data are too sparse to confidently justify analysis of these predictions without additional support in the future. None of the genes that we tested that affect individual life history traits also affected fly fitness, suggesting that additional interrogation is necessary to identify candidate bacterial genes that influence fly fitness.

In summary, our current work identifies bacterial strain-specific influences on reproductive timing and output and on overall fly fitness in a laboratory setting. The presented data support and extend a previous analysis of the impact of individual bacterial and microbial communities on the same phenotypes. They also provide an important context for considering the roles of the microbiota in the natural life history and in local adaptation of wild fly populations that naturally bear distinct communities

Microbiology Spectrum

**TABLE 2** Bacterial strains used in this study

| Identifier | Relevant characteristics | Medium | Oxygen condition | Accession no.[a] | Reference[b] |
|---|---|---|---|---|---|
| 7636 | *Escherichia coli* BW25113, CGSC wild-type | LB | Oxic | | 61, 62 |
| 10862 | CGSC#7636 Δ*metH*786::*kan*; KmR | LB | Oxic | | 61, 62 |
| aanb | AAB; *Acetobacter aceti* NBRC 14818 | mMRS | Oxic | BABW00000000 | |
| apan | AAB; *Acetobacter pasteurianus* NBRC 101655 | mMRS | Oxic | NZ_AP014881 | |
| apoc | AAB; *Acetobacter pomorum* DmCS_004 | mMRS | Oxic | JOKL00000000 | |
| atrc | AAB; *Acetobacter tropicalis* DmCS_006 | mMRS | Oxic | JOKM00000000 | |
| atrn | AAB; *Acetobacter tropicalis* NBRC 101654 | mMRS | Oxic | BABS00000000 | |
| bsub | *Bacillus subtilis* subsp. subtilis strain168 | LB | Oxic | NC_000964.3 | |
| ecok | *Escherichia coli* strain K-12 substr. MG1655 | LB | Oxic | NC_000913.3 | |
| lbga | LAB; *Lactobacillus brevis* subsp. gravesensis ATCC 27305 | mMRS | Microoxic | ACGG00000000 | |
| lbrc | LAB; *Lactobacillus brevis* DmCS_003 | mMRS | Microoxic | JOKA00000000 | |
| lfrc | LAB; *Lactobacillus fructivorans* DmCS_002 | mMRS | Microoxic | JOJZ00000000 | |
| lfrk | LAB; *Lactobacillus fructivorans* KCTC 3543 | mMRS | Microoxic | AEQY00000000 | |
| lplc | LAB; *Lactiplantibacillus plantarum* DmCS_001 | mMRS | Microoxic | JOJT00000000 | |
| lplw | LAB; *Lactobacillus plantarum* WCFS1 | mMRS | Microoxic | NC_004567.2 | |
| pput | *Pseudomonas putida* F1 | LB | Oxic | NC_009512.1 | |
| Pvec | AAB; *Acetobacter pasteurianus* 3p3 (apa3, 39) +empty pCM62 | mMRS | Oxic | | 22 |
| Sdr | AAB; *Acetobacter pasteurianus* 3p3 (apa3, 39) +pCM62-SDR | mMRS | Oxic | | 22 |
| Gdh | AAB; *Acetobacter pasteurianus* 3p3 (apa3, 39) +pCM62-GDH | mMRS | Oxic | | 22 |
| Gndh | AAB; *Acetobacter pasteurianus* 3p3 (apa3, 39) +pCM62-GnDH | mMRS | Oxic | | 22 |

[a]Accession numbers are provided for strains used in the MGWA.
[b]References are provided for mutants created in previous studies.

of microorganisms. We anticipate that future work investigating the bacterial genetic basis for shaping fly fitness will help us understand the current evolved relationship between hosts and their microbial partners in natural, wild settings.

## MATERIALS AND METHODS

**Fly and bacterial culture.** All experiments were conducted using a *D. melanogaster* CantonS line originally obtained from Mariana Wolfner that is free of the endosymbiont *Wolbachia*. Standard growth conditions were at 25°C on a 12-h light/dark cycle at ambient (~25 to 40%) humidity. Flies in standard culture were reared on a yeast-glucose (Y-G) diet (10% brewer's yeast, 10% glucose, 1% agar, 0.0415% phosphoric acid, and 0.415% propionic acid). Bacterial strains were cultured on the medium and oxygen conditions listed in Table 2. The strains were selected because they were AAB, LAB, or enterobacteria that were isolated from flies (strain codes apoc, atrc, lbrc, lfrc, lplc) or not (aanb, apan, atrn, lbga, lfrk, lplw, ecok, pput) or because they are common lab bacteria that are not normally found abundantly in flies (bsub, ecok). The medium was either Miller lysogeny broth (LB) (Genesee Scientific; no. 11-122) or modified de Man-Rogosa-Sharpe medium (mMRS) (41). Oxygen conditions were either oxic (ambient oxygen, shaking for liquid culture) or microoxic ($CO_2$-flooded, sealed container for solid culture, static for liquid culture). The accession numbers in Table 2 correspond to the exact strains that we tested.

To control the microbiota composition of the flies, we dechorionated eggs and either left the eggs undisturbed or introduced bacteria in pure culture as in our previous work, where the techniques were successful (42). Briefly, *Drosophila* embryos were collected at an age of <20 h, dechorionated in two 150-s washes with 0.6% sodium hypochlorite, rinsed three times with sterile water, and transferred in batches of 30 to 60 eggs with a paintbrush to 7.5 ml of sterile diet in a 50-ml polypropylene centrifuge tube. The sterile diet was the Y-G diet, omitting the propionic and phosphoric acid. Axenic flies were prepared by leaving the transferred embryos undisturbed. Monoassociated flies were derived by separately culturing the bacterium of interest overnight, washing it in phosphate-buffered saline (PBS), normalizing it to an optical density at 600 nm ($OD_{600}$) of 0.1 in PBS, and inoculating 50 $\mu$l of normalized culture to the sterile diet within 12 h of egg transfer. We reared gnotobiotic 5-species flies following established procedures (41). Cultures of *L. plantarum* DmCS_001, *Lactobacillus fructivorans* DmCS_002, *L. brevis* DmCS_003, *Acetobacter pomorum* DmCS_004, and *A. tropicalis* DmCS_006 were grown, washed, normalized as above, and then mixed in equal ratios, and 50 $\mu$l of the bacterial mix was added to *D. melanogaster* eggs on the diet. In each experiment, we confirmed the success of the axenic and mono- or polyassociations by dilution plating homogenates of representative pools of flies to confirm that the adult flies bore the intended microbiota composition.

**Fecundity.** *D. melanogaster* fecundity was defined as the number of F1 offspring that reached pupation and was measured twice-weekly over a 4-week interval. First, 30 to 60 P generation *D. melanogaster* specimens per vial were monoassociated with different bacterial strains, a 5-species mix, or were left axenic. After >90% of the P generation had eclosed, an 18-h fecundity measure was prepared by transferring P flies to a bacteria-seeded Y-G diet (no preservative) between 8 and 10 h into the daily light cycle. The vials were preincubated with 50 $\mu$l $OD_{600}$ = 0.1 normalized bacteria (in each case the same treatment as originally) to ensure the flies did not lose exposure to bacteria because of vial transfers. Bacteria were prewashed and resuspended in PBS to avoid transfer to the diet of medium nutrients.

**A**

| Age class | $D^A$ | $O^B$ | $T^C$ | $A^D$ | $f^E$ | $s^F$ |
|---|---|---|---|---|---|---|
| 0 | 0 | 191 | 18.024 | 14 | 0 | 1 |
| 1 | 12 | 191 | 18.024 | 14 | 54.5 | 1 |
| 2 | 15 | 212 | 18.216 | 14 | 79.8 | 1 |
| 3 | 19 | 500 | 18.36 | 14 | 140 | 0.93 |
| 4 | 22 | 91 | 18.072 | 13 | 37.2 | 0.46 |
| 5 | 26 | 22 | 20.016 | 6 | 13.2 | 1 |
| 6 | 29 | 5 | 18.168 | 6 | 4.4 | 1 |
| 7 | 33 | 1 | 17.928 | 6 | 0.67 | 1 |
| 8 | 36 | 0 | 18.552 | 6 | 0 | 1 |
| 9 | 37 | 0 | 18.552 | 6 | 0 | na |

**B**

| | $f^0$ | $f^1$ | $f^2$ | $f^3$ | $f^4$ | $f^5$ | $f^6$ | $f^7$ | $f^8$ | $f^9$ |
|---|---|---|---|---|---|---|---|---|---|---|
| *f values* | 0 | 54.5 | 79.8 | 140.0 | 37.2 | 13.2 | 4.4 | 0.67 | 0 | 0 |
| $s^0$ | 1 | 0 | 0 | 0 | 0 | 0 | 0 | 0 | 0 | 0 |
| $s^1$ | 0 | 1 | 0 | 0 | 0 | 0 | 0 | 0 | 0 | 0 |
| $s^2$ | 0 | 0 | 1 | 0 | 0 | 0 | 0 | 0 | 0 | 0 |
| $s^3$ | 0 | 0 | 0 | 0.93 | 0 | 0 | 0 | 0 | 0 | 0 |
| $s^4$ | 0 | 0 | 0 | 0 | 0.46 | 0 | 0 | 0 | 0 | 0 |
| $s^5$ | 0 | 0 | 0 | 0 | 0 | 1 | 0 | 0 | 0 | 0 |
| $s^6$ | 0 | 0 | 0 | 0 | 0 | 0 | 1 | 0 | 0 | 0 |
| $s^7$ | 0 | 0 | 0 | 0 | 0 | 0 | 0 | 1 | 0 | 0 |
| $s^8$ | 0 | 0 | 0 | 0 | 0 | 0 | 0 | 0 | 1 | 0 |

$^A$Day
$^B$# of F1s detected. If too many to count we assigned it the value '500', a value greater than the largest number we could count in any vial
$^C$Time laying F1 eggs (h)
$^D$Females alive at period start
$^E(O_x / (T_x * A_x)) * (24 * (D_{x+1} - D_x))$
$^F s = A_{x+1}/A_x$

**FIG 4** Leslie matrix construction. Sample construction of a Leslie matrix from raw data is shown for one vial ('11c-13', or replicate vial 3 [of 3] for *A. tropicalis* DmCS_006 on experimental day 1 [of 3]).

Eighteen hours later (2 to 4 h into the daily light cycle the following day), the P generation was transferred to a sterile Y-G diet (no bacterial seeding) for 2 to 3 days until the next cycle of 18-h fecundity measures was initiated. This process was performed twice a week for 4 weeks, providing 8 measures of fecundity. The spent 18-h vials were stored at 20°C on an uncontrolled light cycle, approximately 6 a.m. to 6 p.m., and the number of pupae that formed after approximately 2 weeks was counted and normalized to the number of P generation females in the vial at the beginning of each egg-laying interval. Potential contamination during transfers was monitored weekly by homogenizing a pool of five F1 progeny that emerged from spent vials for each P generation vial. We only tested for contamination in vials that contained sterile Y-G diet and not those preseeded with bacteria. If the F1 vials contained ≥200 CFU fly$^{-1}$ of an unexpected bacterial colony morphology in two consecutive weeks, that vial of P flies was removed from all subsequent analyses. Three separate experiments, each with triplicate vials, were performed on three consecutive days, and vials for all experiments were transferred on the same days each week, i.e., fecundity was measured on the same day, but the flies in different vials were up to 2 days apart in age, depending on which experiment they began in. Significant differences in fly fecundity were calculated using a Kruskal-Wallis test, and significant differences between treatments were performed using a Benjamini-Hochberg (B-H) corrected Dunn test (43) and shown as compact letter displays (44). Means from multiple experiments were calculated based on date of vial transfers. A nonredundant analysis of data from the first age class was presented in Fig. 1 of a previous publication (14).

**Fly life span.** At each vial transfer in the fecundity experiment, natural fly mortality was assessed by recording the number and sex of the dead flies that remained in the spent vials. At the experiment's conclusion, all surviving flies were anesthetized to determine their number and sex. All data for male flies were discarded because the fecundity data only correspond to female flies. At each time point, when a fly died or was lost during vial transfers, the time from egg collection and either a 1 or a 0, respectively, was recorded. All flies that were alive at the end of the experiment were recorded as "0" at the time the experiment concluded. Significant differences between treatments were determined by a log rank test in R. Compact letter displays to display significant differences between treatments with B-H correction were determined using the survminer (45) and multcomp (46) packages. Survival curves were created in R (47).

**Fly fitness.** From the fecundity and life span data, we calculated the fitness of flies in each vial (except for vials discarded for contamination, triplicate vials in each of three separate experiments for a total of 9 vials). Fitness was calculated from the twice-weekly fecundity data paired with fly life span measurements over the same period so that Leslie matrices (23, 48) could be used to calculate maximum rates of population growth, a standard measure of fitness. Thus, each age class spanned 3 or 4 days, and values for the full window were extrapolated from measurements during its first 18 h. Values were from the data collected as reported above and were from the time that flies hatched until most treatments no longer laid viable eggs. We constructed the Leslie matrices by assigning age classes as the twice weekly intervals when fly fecundity was measured, and then deriving s, the fractional survival of individuals between age classes, and f, the number of offspring produced per female at each age class (Fig. 4). In sequence, the value f was listed across the top of the matrix and s across the diagonal of the matrix for each time interval. The first and last data points were manually added with assumptions. First, we assumed 100% survival for the P generation eggs from egg-laying until the first age class. That is, if 20 female flies were calculated at the end of the experiment, we assumed that 20 female eggs were transferred to the fly diet at the beginning of the experiment. Second, we assigned a 0 value for f to flies from egg picking until the first vial transfers began. Third, we assigned a final 0 f value to flies at the conclusion of the experiment, even if fecundity had not completely been exhausted, so that the matrices did not reflect never-ending, low-level reproduction. Although this last assumption could hypothetically underestimate the fitness of strains that were still reproductively active at the conclusion of the experiment, the relatively small f values at late versus early age classes suggest that this assumption is unlikely to dramatically influence the fitness values. This idea is supported by results in Fig. S1 and S2 in the

supplemental material, which show that permuting survival and fecundity of late-stage age classes had a negligible influence on fitness. Finally, we calculated the eigenvector of the matrix in R using the "eigen" function and extracted the first eigenvalue as the vial's fitness value. Then, using the first eigenvalue as a response variable, a Kruskal-Wallis rank sum test was used to test for treatment-level differences in fecundity. If the Kruskal-Wallis test was significant, then differences between treatments were assessed by a Dunn's multiple comparison test (43), and differences between treatments were shown as compact letter displays (44).

**Fitness permutations.** Fitness permutation classes were calculated by manually adjusting the survival or fecundity values collected in the previously described experiments. We began with the Leslie matrices that were constructed for each vial and for each age class in the Leslie matrix and permuted either its fecundity or its survival value to a specified percentage of its observed value before recalculating fitness (i.e., the eigenvalue of the Leslie matrix). Differences between the fitness values of the original and permuted Leslie matrices were calculated by a Kruskal-Wallis test, which, if significant, was followed by a B-H corrected Dunn test. The correlations between permuted values were tested by a Spearman rank test. *P. putida* data were excluded from the permutational analyses because the survival and fecundity data indicated *P. putida* acted more as a pathogen than a member of the healthy fly microbiota (see Fig. 1A and C). Calculated fitness values are in Table S4 in the supplemental material.

**Metagenome-wide association.** We performed a metagenome-wide association to identify the association between bacterial genes and the variation in fly fitness as in our previous work (49). First, we obtained genome sequences from public databases for the exact strains used in our study (Table 2). Then, we clustered the amino acid sequences from each genome into orthologous groups (OGs) using the OrthoMCL software with an inflation factor of 1.5. The response variable was the fitness values, which were normally distributed (Shapiro test $P > 0.05$) if the fitness conferred by *P. putida* was omitted from the analyses. We proceeded with *P. putida* fitness values excluded because *P. putida* appeared to act more as a pathogen than a member of the normal fly microbiota. Finally, we assigned a significance value to each phylogenetic distribution group, defined as the exact set of strains in which an OG was present, using a linear mixed effects model, with OG presence as the main effect and experimental replicate as a random effect. The *P* values were Bonferroni corrected. Together, these approaches reported the significance of the association between bacterial OG presence/absence and fly fitness. The strain code names and the raw MGWA results can be found in Tables S5 and S3 in the supplemental material, respectively.

**Statistics and data analysis.** Statistics and data analysis were performed in R and are shown in File S1 in the supplemental material. Packages that were not cited elsewhere are included (50–60). Raw, executed code is included as Data Set S1 in the supplemental material. Unexecuted code and raw data files can be accessed on github by pasting the following into an R window: devtools::install_github("john-chaston/df3") and learnr::run_tutorial("lesson4","drosfitness"). devtools, learnr, and packages used in our work must be installed.

## SUPPLEMENTAL MATERIAL

Supplemental material is available online only.

**SUPPLEMENTAL FILE 1**, PDF file, 1.8 MB.

**SUPPLEMENTAL FILE 2**, XLSX file, 1.9 MB.

## ACKNOWLEDGMENTS

We thank three anonymous reviewers for comments that led to important improvements in the manuscript.

This work and its publication were supported by startup funds from Brigham Young University and the National Institute of General Medical Sciences of the National Institutes of Health under award number R15GM140388 to J.M.C.

The content is solely the responsibility of the authors and does not necessarily represent the official views of the National Institutes of Health.

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
