## [Reviewer comments · Microbiology Spectrum]

Microbiology Spectrum

The microbiota influences fitness and the timing of reproduction in the fruit fly *Drosophila melanogaster*

Melinda Matthews, Jaanna Malcolm, and John Chaston

Corresponding Author(s): John Chaston, Brigham Young University

Review Timeline:

Submission Date:	April 13, 2021
Editorial Decision:	May 29, 2021
Revision Received:	July 30, 2021
Accepted:	August 21, 2021

Editor: Silvia Bulgheresi

Reviewer(s): Disclosure of reviewer identity is with reference to reviewer comments included in decision letter(s). The following individuals involved in review of your submission have agreed to reveal their identity: William Ludington (Reviewer #1)

Transaction Report:

DOI: <https://doi.org/10.1128/Spectrum.00034-21>

May 29, 2021

Dr. John M Chaston
Brigham Young University
Plant & Wildlife Sciences
701 E University Pkwy
4105 LSB
Provo, UT 84602

Re: Spectrum00034-21 (The microbiota influences fitness and the timing of reproduction in the fruit fly *Drosophila melanogaster*)

Dear Dr. John M Chaston:

Thank you for submitting your manuscript to Microbiology Spectrum. As you will see the reviewers support publication of a revised paper. Please revise the paper along the lines suggested by the reviewers. When submitting the revised version of your paper, please provide (1) point-by-point responses to the issues raised by the reviewers as file type "Response to Reviewers," not in your cover letter, and (2) a PDF file that indicates the changes from the original submission (by highlighting or underlining the changes) as file type "Marked Up Manuscript - For Review Only". Please use this link to submit your revised manuscript - we strongly recommend that you submit your paper within the next 60 days or reach out to me. Detailed information on submitting your revised paper are below.

Link Not Available

Sincerely,

Silvia Bulgheresi

Journals Department
Reviewer comments:

Reviewer #1 (Comments for the Author):

I remain enthusiastic about the work. All of my comments have been sufficiently addressed. I appreciate the other reviewers' comments. While adding complexity to the microbiome combinations would be an improvement, I feel the present work is a necessary first step and sufficient to justify publication.

The major strength of this paper is in properly defining and analyzing fitness under different microbiome contexts. While many animal studies use physiology metrics to infer fitness (e.g. lifespan, lipid content, egg laying), the present work makes the actual calculations. This is important because the physiology metrics are often tradeoffs, so their effects on organismal fitness must be calculated using appropriate models. The authors have carefully done so here, and I feel this work may therefore become a standard in the field. The provision of the R code to make the calculations is a valuable resource.

Lastly, just a clarification to a previous comment I made. The work I was referring to in my comment that starts with "line 169" is work (i) out of the Lee and Leulier labs with respect to individual genes and (ii) out of the Ja lab with respect to bulk nutrition. I think it is not necessary to add citations.

Reviewer #2 (Comments for the Author):

Matthews et al describe the role of the microbiota on reproduction and survival of *Drosophila melanogaster*, with an interesting computer modeling comparison of the influence of survival rate on fitness. While they did not identify genes that are associated with variation in fly fitness, the breadth of strains tested and the potential interaction between survival and fitness is intriguing. The writing is well done, the experimental design and implementation are commendable, and the integration of ecological modeling in flies adds strength to the paper. I do agree with a previous reviewer that many of the findings are not novel, however, there are some new twists on previous studies that are important to the field. While I cannot assess the mathematical modeling, I do have minor comments to clarify how the experiments were done and the interpretation.

Major:

Lines 83-84: I was confused by the reference to flies at different latitudes. It wasn't until I read the discussion that I understood that this might be due to relative abundance of AAB:LAB. Please clarify that here.

Line 87-88: I am confused by the statement "Because there are no methods for colonizing flies with the same microbial communities in variable ratios...." Isn't that what reference 21 describes? Later in the text (line 288 alludes to this) it appears that the issue is "stable abundances" of the ratio of microbes. Does this mean that this experiment included generations of flies exposed to the individual or polymicrobial communities? From the methods, I thought that there were 8 replicates of the experimental exposures done in triplicate, and that each replication started with a new generation derived germ free. The comment about "stable communities" makes me wonder if this is a multigenerational study and I misinterpreted this. If this is not the case, please clarify. Does this also mean that the mix of 5 microbes in this manuscript is not a stable community?

Line 89, similar to above, describing 1:0 as a hypothetical extrema is a bit of a stretch. I realize that this is in response to an earlier reviewers comments. I think there is value in understanding how an individual strain of microbe affects the host before examining how it affects the host when in combination (see Rolig et al 2015). Another way to address this is to talk more about the mixed community of 5 members (described in line 356-260) and whether the effect the individual microbes had on host fitness and survival predicted the outcome in the 5 member exposure. That may be beyond the scope of this project.

Minor:

Line 70: add "of" between "comprises" and "acetic acid"

Line 106: Explain why those specific strains were selected for testing.

The 5 member mixed community is not referred to in the text, as far as I can tell. It's not completely clear why the members of that specific community were chosen.

Table 2 and figure 1: It's not clear which are AAB vs LAB. The color key is given in figure 2 (I appreciate keeping the same color key throughout). Include the AAB/LAB information in table 2 or in the legend in figure 1.

Line 144: Specify "in monoassociation" when describing that AAB are more beneficial to flies than LAB.

Section starting with line 146: It is not clear whether these results were from an experiment or computational modeling until the line 177. State that early, potentially in line 153: "We investigated this idea by simulating changes in reproduction and survival in the flies using computational modeling..."

Line 187: Clarify that the strains overexpress the genes and add a reference at the end this sentence. Also clarify in lines 191-192 whether these are in bacteria with overexpressing genes in a wildtype background or a knockout (it appears that these are overexpressed genes, but it is not clear). These strains are described in table 2, so reference that here with clarification of which strains in table 2 were used for this experiment.

Line 213: "after correcting for multiple tests" - describe the tests briefly in the methods.

Line 397/398: why were males discarded? I've seen this before in fly studies, but it's not clear why that data is discarded here.

Line 450-451: It is unclear whether the fitness values are the response variables or something different. Clarify with "and" or "such as" between "values," and "the fitness".

Table 1: column "taxa contain" - there are several LAB strains and AAB strain, which contain the genes in the last column?

Reviewer #3 (Comments for the Author):

The manuscript entitled "The microbiota influences fitness and the timing of reproduction in the fruit fly *Drosophila*" describes a large scale experiment to identify the phenotypic effects of single bacterial symbionts on life history traits of the *Drosophila* host and resulting fitness differences. As already mentioned by the previous reviewers, this is an interesting and comprehensive, but not entirely novel experiment, which confirms and extends previous analyses.

After carefully reading the revised manuscript and the responses to three previous reviewers, I feel that this manuscript would benefit from further clarifications and streamlining. I find the manuscript often too speculative and, particularly in the results section, I am missing sufficient details about statistical analyses and actual results.

Similar to previous reviewer 2, I would recommend better definitions of "fitness" per se and of the life history traits that were investigated as proxies for fitness. The Leslie matrix is used to estimate growth rates, which is a direct fitness measure. Conversely, when the authors talk of an individual's fitness, they rather refer to values of fitness-related traits and NOT a direct measure of fitness. Moreover, I would consistently use the term "fecundity" (which is actually measured as the number of offspring) rather than "reproduction", which describes the biological process of producing offspring. Alternatively (as also suggested by reviewer 2), you could use "reproductive rate". Moreover, I would rather use the terms "longevity" or "lifespan" than "survival". "Survival" refers to the propensity of surviving given a condition or age, while you are actually measuring the age at death.

Consistent with the critique from reviewer 2, I am missing summary statistics, value ranges, p-values and degrees of freedom etc. in the entire results section of the revised manuscript. In addition, there are no tables that summarize the results from the statistical analyses, such as for the log-rank-tests or the Kruskal-Wallis tests. Such numerical results along verbal arguments are absolutely necessary and needed for context and comparison.

I am missing a more detailed justification of the bacteria used for the experiments. Are these common species in the gut microbiome, or are they known pathogens? While this information can be found elsewhere, it would be helpful for the reader to get more context in the Materials and Methods.

What about strain-specific differences of bacterial titers in the gut? Weak phenotypic effects in the host may not necessarily be indicative for strain-specific fitness effects but rather the result of varying titer levels. While I understand that it is not possible to generate these additional data, it may be worth to at least discuss such potential problems.

I am missing an in-depth description of the actual permutations in the Material and Methods. You need to better explain what was actually done and which data were used for permutations. In line 23, you mention simulations, does this refer to these permutations or did you carry out other model-based simulations?

Line 48. The sentence starting with „Our results are consistent..." appears to be incomplete.

Throughout the MS consistently use "trade-off" rather than "tradeoff"

Line 75. The revised sentence is not entirely clear and may benefit from a rework. What do you mean with positions? Positions along the continuum?

Line 108. Rather than writing "we measured the fitness of flies", I would say "we measured fecundity and longevity as proxies for fitness".

Line 153. Change "we detected the different ..." to "we detected THAT the different ..."

Lines 179. What exact age class do you mean by the youngest? It would be very useful to be more explicit in the text. How were these tested? was this significant? While this is mentioned in the M&M, it would be helpful to add more details here.

Staff Comments:

Preparing Revision Guidelines

For complete guidelines on revision requirements, please see the Instructions to Authors at [link to page]. **Submissions of a paper that does not conform to Microbiology Spectrum guidelines will delay acceptance of your manuscript.**

Please return the manuscript within 60 days; if you cannot complete the modification within this time period, please contact me. If you do not wish to modify the manuscript and prefer to submit it to another journal, please notify me of your decision immediately so that the manuscript may be formally withdrawn from consideration by Microbiology Spectrum.

If you would like to submit an image for consideration as the Featured Image for an issue, please contact Spectrum staff.

Corresponding authors may join or renew ASM membership to obtain discounts on publication fees. Need to upgrade your membership level? Please contact Customer Service at

Service@asmusa.org.

Reviewer #1 (Comments for the Author):

I remain enthusiastic about the work. All of my comments have been sufficiently addressed. I appreciate the other reviewers' comments. While adding complexity to the microbiome combinations would be an improvement, I feel the present work is a necessary first step and sufficient to justify publication.

The major strength of this paper is in properly defining and analyzing fitness under different microbiome contexts. While many animal studies use physiology metrics to infer fitness (e.g. lifespan, lipid content, egg laying), the present work makes the actual calculations. This is important because the physiology metrics are often tradeoffs, so their effects on organismal fitness must be calculated using appropriate models. The authors have carefully done so here, and I feel this work may therefore become a standard in the field. The provision of the R code to make the calculations is a valuable resource.

Lastly, just a clarification to a previous comment I made. The work I was referring to in my comment that starts with "line 169" is work (i) out of the Lee and Leulier labs with respect to individual genes and (ii) out of the Ja lab with respect to bulk nutrition. I think it is not necessary to add citations.

We appreciate the reviewer's time, insight, and clarification. We did not make any changes in the manuscript.

Reviewer #2 (Comments for the Author):

Matthews et al describe the role of the microbiota on reproduction and survival of *Drosophila melanogaster*, with an interesting computer modeling comparison of the influence of survival rate on fitness. While they did not identify genes that are associated with variation in fly fitness, the breadth of strains tested and the potential interaction between survival and fitness is intriguing. The writing is well done, the experimental design and implementation are commendable, and the integration of ecological modeling in flies adds strength to the paper. I do agree with a previous reviewer that many of the findings are not novel, however, there are some new twists on previous studies that are important to the field. While I cannot assess the mathematical modeling, I do have minor comments to clarify how the experiments were done and the interpretation.

Major:

Lines 83-84: I was confused by the reference to flies at different latitudes. It wasn't until I read the discussion that I understood that this might be due to relative abundance of AAB:LAB. Please clarify that here.

We modified the text at L84-90 to address this concern. It now reads:

"This goal was motivated by our detection of a geographic pattern in the microbiota composition of wild flies, where AAB and LAB were more abundant in flies at lower and higher latitudes, respectively, in multiple locations; and work showing that such

variation in microbiota composition was likely linked to the adaptive traits of the flies (14). However, the previous work did not address variation in fly fitness."

Line 87-88: I am confused by the statement "Because there are no methods for colonizing flies with the same microbial communities in variable ratios...." Isn't that what reference 21 describes? Later in the text (line 288 alludes to this) it appears that the issue is "stable abundances" of the ratio of microbes. Does this mean that this experiment included generations of flies exposed to the individual or polymicrobial communities? From the methods, I thought that there were 8 replicates of the experimental exposures done in triplicate, and that each replication started with a new generation derived germ free. The comment about "stable communities" makes me wonder if this is a multigenerational study and I misinterpreted this. If this is not the case, please clarify. Does this also mean that the mix of 5 microbes in this manuscript is not a stable community?

We removed this line to be more correct and to avoid confusion. We intended to explain that rearing gnotobiotic flies leads to a stable microbial composition regardless of the ratios the microbes are combined in (shown in Newell and Douglas, AEM, 2014); therefore, we can't compare 10%AAB-90%LAB flies with 90%AAB-10%LAB flies. However, the text is unclear and overstates the barrier because Newell does show some variation can be obtained by mixing very low amounts of bacteria (but issues remain with outliers that don't really make this a reasonable approach, which was the point we were communicating). We judge that the followup sentence gives sufficient context without the confusion (L90-92):

"For congruence with that previous work and to understand the roles of individual microbes, we used mono-associated animals for these experiments, and asked three questions:"

Line 89, similar to above, describing 1:0 as a hypothetical extrema is a bit of a stretch. I realize that this is in response to an earlier reviewers comments. I think there is value in understanding how an individual strain of microbe affects the host before examining how it affects the host when in combination (see Rolig et al 2015). Another way to address this is to talk more about the mixed community of 5 members (described in line 356-260) and whether the effect the individual microbes had on host fitness and survival predicted the outcome in the 5 member exposure. That may be beyond the scope of this project.

We agree with the reviewer on the benefits of understanding monoassociations first, so we replaced this line with this statement (now L90-91):

"For congruence with that previous work and to understand the roles of individual microbes, we used mono-associated animals for these experiments..."

Minor:

Line 70: add "of" between "comprises" and "acetic acid"

Because a previous reviewer did not like 'comprised of' we rephrased to *"the most common and abundant microbes are the acetic acid..."* (now L71)

Line 106: Explain why those specific strains were selected for testing.

We explained why the strains were selected by adding text at L 96-8 and 354-8:

96-8: *"The strains we used were selected to obtain a broad representation of AAB and LAB, with several additional strains to add genetic diversity to the strain panel."*

354-8: *The strains were selected because they were AAB, LAB, or enterobacteria that were isolated from flies (strain codes apoc, atrc, lbrc, lfrc, lplc) or not (aanb, apan, atrn, lbga, lfrk, lplw, ecok, pput), or because it is a common lab bacterium that is not normally found abundantly in flies (bsub; ecok).*

The 5 member mixed community is not referred to in the text, as far as I can tell. It's not completely clear why the members of that specific community were chosen.

We describe the composition of the 5 member community at L 365-7. In revision we added text at L373-6 to explain that they were chosen because they were established in previous work.

The section now reads as:

"We reared gnotobiotic 5-species flies following established procedures (Newell, 2014). Cultures of L. plantarum DmCS_001, L. fructivorans DmCS_002, L. brevis DmCS_003, A. pomorum DmCS_004, and A. tropicalis DmCS_006 were grown..."

Table 2 and figure 1: It's not clear which are AAB vs LAB. The color key is given in figure 2 (I appreciate keeping the same color key throughout). Include the AAB/LAB information in table 2 or in the legend in figure 1.

We added the color descriptions to the legend for Figure 1 (L683-5) and added AAB and LAB designations to Table 2.

"Colors are according to the high-level taxonomic classification of the microbial treatment: red (AAB), blue (LAB), purple (B. subtilis), green (Enterobacteriales), magenta (5-sp), black (axenic)."

Line 144: Specify "in monoassociation" when describing that AAB are more beneficial to flies than LAB.

We made this change (now L145):

"Another similarity between our work and Gould et al. (21) was that in mono-association the AAB tended to confer higher fitness values on flies than LAB (21)."

Section starting with line 146: It is not clear whether these results were from an experiment or computational modeling until the line 177. State that early, potentially in line 153: "We investigated this idea by simulating changes in reproduction and survival in the flies using computational modeling..."

We made this change exactly as recommended (now L158).

Line 187: Clarify that the strains overexpress the genes and add a reference at the end this sentence. Also clarify in lines 191-192 whether these are in bacteria with overexpressing genes in a wildtype background or a knockout (it appears that these are overexpressed genes, but it is not clear). These strains are described in table 2, so reference that here with clarification of which strains in table 2 were used for this experiment.

We modified the text at L195-8, which now reads as:

“For example, in Acetobacter pasteurianus 3p3 overexpression of S-oxidoreductase, gluconate dehydrogenase, and glucose dehydrogenase genes led to increased catabolism of dietary glucose by the bacteria and lowered availability of dietary glucose for fly fat storage ((22), strains Sdr, Gndh, and Gdh from Table 2, respectively).”

Line 213: "after correcting for multiple tests" - describe the tests briefly in the methods.

The tests are described in the methods with the following text at L477-80:

“Finally, we assigned a significance value to each phylogenetic distribution group, defined as the exact set of strains in which an OG was present, using a linear mixed effects model, with OG presence as the main effect and experimental replicate as a random effect. The p-values were Bonferroni corrected.”

Line 397/398: why were males discarded? I've seen this before in fly studies, but it's not clear why that data is discarded here.

We added text at L415 to explain that male flies were discarded because the fecundity data only correspond to female flies. Fly studies sometimes study just one sex to limit the number samples that must be prepared or because a trait only applies to one sex, although that was not the reason in this case. The final text reads as:

“All data for male flies was discarded because the fecundity data only correspond to female flies.”

Line 450-451: It is unclear whether the fitness values are the response variables or something different. Clarify with "and" or "such as" between "values, " and "the fitness".

We modified the text at L473-5, which now reads as:

“The response variable was the fitness values, which were normally distributed...”

Table 1: column "taxa contain" - there are several LAB strains and AAB strain, which contain the genes in the last column?

We changed the table column to be “Strains that contain the PDG”

Reviewer #3 (Comments for the Author):

The manuscript entitled "The microbiota influences fitness and the timing of reproduction in the fruit fly *Drosophila*" describes a large scale experiment to identify the phenotypic effects of single bacterial symbionts on life history traits of the *Drosophila* host and resulting fitness differences. As already mentioned by the previous reviewers, this is an interesting and comprehensive, but not entirely novel experiment, which confirms and extends previous analyses.

After carefully reading the revised manuscript and the responses to three previous reviewers, I feel that this manuscript would benefit from further clarifications and streamlining. I find

the manuscript often too speculative and, particularly in the results section, I am missing sufficient details about statistical analyses and actual results.

Similar to previous reviewer 2, I would recommend better definitions of "fitness" per se and of the life history traits that were investigated as proxies for fitness. The Leslie matrix is used to estimate growth rates, which is a direct fitness measure. Conversely, when the authors talk of an individual's fitness, they rather refer to values of fitness-related traits and NOT a direct measure of fitness. Moreover, I would consistently use the term "fecundity" (which is actually measured as the number of offspring) rather than "reproduction", which describes the biological process of producing offspring. Alternatively (as also suggested by reviewer 2), you could use "reproductive rate". Moreover, I would rather use the terms "longevity" or "lifespan" than "survival". "Survival" refers to the propensity of surviving given a condition or age, while you are actually measuring the age at death.

In most places we replaced the word 'reproduction' with the word 'fecundity' (41 replacements were made, we haven't included the line numbers). In some cases we retained use of the word reproduction, such as when speaking in general terms about the trade-off between somatic maintenance and reproduction, or when discussing how Leslie matrices explain the relationship between survival and reproductive output. We also changed survival to lifespan in at least seven place in the text. However, since Leslie matrices are age-structured models where survival between intervals is the key phenotype, we retained the word 'survival' throughout much of the text, especially when we describe the results of the computer simulations. We hope that we have balanced use of these terms correctly and are open to further guidance if the revised descriptions are wrong.

Consistent with the critique from reviewer 2, I am missing summary statistics, value ranges, p-values and degrees of freedom etc. in the entire results section of the revised manuscript. In addition, there are no tables that summarize the results from the statistical analyses, such as for the log-rank-tests or the Kruskal-Wallis tests. Such numerical results along verbal arguments are absolutely necessary and needed for context and comparison.

We sincerely apologize for our mistake. As part of the previous revision we created new figures that contain the statistics for the previous revision, but forgot to replace them during submission. We have corrected this error now. We had previously added Table S2, which includes the statistics for the many comparisons in the log rank tests and this is cited at current lines L124, 207, and 211. If there are additional statistical details that are missing please let us know, but we believe this corrects the issue.

I am missing a more detailed justification of the bacteria used for the experiments. Are these common species in the gut microbiome, or are they known pathogens? While this information can be found elsewhere, it would be helpful for the reader to get more context in the Materials and Methods.

We added text at line 96-8 (suggested by another reviewer) and 354-8 to clarify this issue:

96-8: "The strains we used were selected to obtain a broad representation of AAB and LAB, with several additional strains to add genetic diversity to the strain panel."

354-8: The strains were selected because they were AAB, LAB, or enterobacteria that were isolated from flies (strain codes apoc, atrc, lbrc, lfrc, lplc) or not (aanb, apan, atrn, lbga, lfrk, lplw, ecok, pput), or because it is a common lab bacterium that is not normally found abundantly in flies (bsub; ecok).

What about strain-specific differences of bacterial titers in the gut? Weak phenotypic effects in the host may not necessarily be indicative for strain-specific fitness effects but rather the result of varying titer levels. While I understand that it is not possible to generate these additional data, it may be worth to at least discuss such potential problems.

We added text to mention this issue at L259-62. Gould et al did find some modest influence of bacterial abundance on poly-associated flies (bacterial abundance affected 2 of 20 possible trait correlations). However, because many LAB tend to be less abundant than many AAB in mono-associated flies, bacterial abundance is rarely correlated with traits in mono-associated flies. For example, we previously tested for and failed to detect a link to lifespan in mono-associated flies (Matthews, AEM, 2020).

“Finally, unlike Gould et al., we did not measure bacterial abundance to test how it was related to fitness because we did not collect data for any poly-associated flies; and because bacterial abundance is not linked to lifespan effects in mono-associated flies for the host genotype and bacterial strains in this study (24).” {24 is the Matthews reference}

I am missing an in-depth description of the actual permutations in the Material and Methods. You need to better explain what was actually done and which data were used for permutations. In line 23, you mention simulations, does this refer to these permutations or did you carry out other model-based simulations?

We provided more details at L456-9 on the methods including specifically stating that the survival and fecundity values were permuted and that the data were the original Leslie matrices calculated for the previous experiments. Also, to make it easier to find we separated the text into a new methods section with these explanations. The entire section is at L455-66 and reads as:

Fitness permutations

Fitness permutations class were calculated by manually adjusting the survival or fecundity values collected in the previously described experiments. We began with the Leslie matrices that were constructed for each vial, and, for each age class in the Leslie matrix and permuted either its fecundity or its survival value to a specified percentage of its observed value before re-calculating fitness (i.e. the eigenvalue of the Leslie matrix). Differences between the fitness values of the original and permuted Leslie matrices were calculated by a Kruskal-Wallis test which, if significant, was followed by a B-H corrected Dunn test. The correlations between permuted values were tested by a Spearman rank test. P. putida data were excluded from the permutational analyses because the survival and fecundity data indicated P. putida acted more as a pathogen than a member of the healthy fly microbiota (See Fig 1AC). Calculated fitness values are in Table S4.

We were using the word 'simulation' to refer to the permutations, so we changed the word to 'permutation' at (now) L21. It now reads as:

"Computational permutations showed that the positive fitness effect of investing in reproduction was reversed if fly survival over time was low..."

Line 48. The sentence starting with „Our results are consistent..." appears to be incomplete.

We checked this sentence and we think it is complete. We ask that the reviewer please alert us if something remains missing (perhaps the tracked changes contributed?). It is now L39-40.

"Our results are consistent with a previous report that, in the laboratory, acetic acid bacteria are more beneficial to the flies than many strains of lactic acid bacteria."

Throughout the MS consistently use "trade-off" rather than "tradeoff"

We made this change at all relevant locations (L53, 55, and 60).

Line 75. The revised sentence is not entirely clear and may benefit from a rework. What do you mean with positions? Positions along the continuum?

We changed the sentence at L 63-5, which now reads as:

"The work presented here defines the fitness consequences to the host of microbial influences that change the host's position on the fast-slow continuum"

Line 108. Rather than writing "we measured the fitness of flies", I would say "we measured fecundity and longevity as proxies for fitness".

We made this change at L95-6. It now reads as:

"To answer these questions, we measured fecundity and longevity as proxies for fitness in flies reared from birth with each of 14 different microorganisms."

Line 153. Change "we detected the different ..." to "we detected THAT the different ..."

We made this change (now L140).

Lines 179. What exact age class do you mean by the youngest? It would be very useful to be more explicit in the text. How were these tested? was this significant? While this is mentioned in the M&M, it would be helpful to add more details here.

We added the text below at L159-64 to clarify the age and add some methods to the results:

"For example, to test the contributions of the bacterial impact on fly survival to fitness during the youngest age class, which corresponded to 0-3 day old adults, we a) sequentially simulated reductions in fly survival by factors of 2, 10, 20, 100, 200, 1000, 10,000, and 100,000; b) calculated fitness based on the permuted survival values; and c) compared the fly fitness values conferred by the different microorganisms by a Kruskal-Wallis test (Figure S1)."

August 21, 2021

Dr. John M Chaston
Brigham Young University
Plant & Wildlife Sciences
701 E University Pkwy
4105 LSB
Provo, UT 84602

Re: Spectrum00034-21R1 (The microbiota influences fitness and the timing of reproduction in the fruit fly *Drosophila melanogaster*)

Dear Dr. John M Chaston:

Your manuscript has been accepted, and I am forwarding it to the ASM Journals Department for publication. You will be notified when your proofs are ready to be viewed.

Sincerely,

Silvia Bulgheresi
Editor, Microbiology Spectrum

Journals Department
Supplemental figures: Accept
Supplemental Material: Accept